# Effectiveness, cost-effectiveness and safety of gabapentin versus placebo as an adjunct to multimodal pain regimens in surgical patients: protocol of a placebo controlled randomised controlled trial with blinding (GAP study)

Sarah Baos [iD],[1] Chris A Rogers [iD],[1] Reyad Abbadi [iD],[2] Aiman Alzetani [iD],[3] Gianluca Casali,[2] Nilesh Chauhan [iD],[4] Laura Collett [iD],[1] Lucy Culliford [iD],[1] Samantha E de Jesus [iD],[1] Mark Edwards [iD],[5,6] Nicholas Goddard,[5] Jennifer Lamb,[1] Holly McKeon [iD],[1] Mat Molyneux,[4] Elizabeth A Stokes [iD],[7] Sarah Wordsworth [iD],[7] Ben Gibbison [iD],[4,8] Maria Pufulete [iD][1]

For numbered affiliations see end of article.

**Correspondence to**
Dr Sarah Baos;
sarah.baos@bristol.ac.uk

## ABSTRACT

**Introduction** Gabapentin is an antiepileptic drug currently licensed to treat epilepsy and neuropathic pain but has been used off-label to treat acute postoperative pain. The GAP study will compare the effectiveness, cost-effectiveness and safety of gabapentin as an adjunct to standard multimodal analgesia versus placebo for the management of pain after major surgery.

**Methods and analysis** The GAP study is a multicentre, double-blind, randomised controlled trial in patients aged 18 years and over, undergoing different types of major surgery (cardiac, thoracic or abdominal). Patients will be randomised in a 1:1 ratio to receive either gabapentin (600 mg just before surgery and 600 mg/day for 2 days after surgery) or placebo in addition to usual pain management for each type of surgery. Patients will be followed up daily until hospital discharge and then at 4 weeks and 4 months after surgery. The primary outcome is length of hospital stay following surgery. Secondary outcomes include pain, total opioid use, adverse health events, health related quality of life and costs.

**Ethics and dissemination** This study has been approved by the Research Ethics Committee . Findings will be shared with participating hospitals and disseminated to the academic community through peer-reviewed publications and presentation at national and international meetings. Patients will be informed of the results through patient organisations and participant newsletters.

**Trial registration number** ISRCTN63614165.

## Strengths and limitations of this study

► Pragmatic design integrated into standard care pathways.
► First trial to assess the impact of gabapentin on hospital stay and quality of life.
► Implemented in three types of major surgery: cardiac, thoracic and abdominal.
► Non-variable dose and limited duration of intervention may reduce applicability (eg, frail/infirm patients and patients requiring analgesia for longer than 2 days).
► Only includes major body cavity surgery, which reduces applicability to major orthopaedic surgery.

## INTRODUCTION

About 4.7 million patients undergo surgery in the UK each year.[1] Many patients experience significant pain after surgery and about 10% experience severe pain.[2–5] Inadequate pain management increases the length of hospital stay[6] and contributes to the development of chronic or persistent post-surgical pain,[7 8] which impacts on quality of life.[9] Current multimodal analgesic regimens include paracetamol, non-steroidal anti-inflammatory drugs, regional analgesia (focused delivery of a local anaesthetic to a specific part of the body) and opioids.[10]

Opioids are key analgesic agents for managing moderate to severe pain. However, they have poor efficacy in movement-associated pain and have significant side-effects including confusion, nausea, vomiting, itching, constipation and respiratory depression. These side effects can increase the length

of hospital stay, delay overall recovery and reduce quality of life.[9] Reliance on opioids after surgery also increases the risk of long-term use and opioid dependence.[11 12] Gabapentin is an antiepileptic drug currently licensed to treat epileptic convulsions and neuropathic pain. It is also commonly used off-label in the perioperative setting to reduce opioid use without compromising pain control. We conducted a survey of UK practice among consultant anaesthetists in the South West of England and members of the British Pain Society Acute Pain Special Interest Group. We found that 35/145 (23%) of anaesthetists prescribe gabapentin to their patients, with large variation in practice across the UK.[13] Reducing opioid use after surgery to return patients to full health as quickly as possible, is one of the central tenets of enhanced recovery in the National Health Service (NHS).[14] However, there is currently no robust evidence to recommend the inclusion of gabapentin in enhanced recovery protocols.

There are over 130 randomised controlled trials (RCTs) that have investigated gabapentin versus placebo in different surgical populations. Most of these trials are small (<200 patients, median 80) and highly heterogeneous, both statistically and clinically. These RCTs have been included in 18 systematic reviews that aimed to assess the effectiveness of gabapentin versus placebo in the perioperative period; 11 of these in surgical populations.[15–25] All reviews reached the same conclusions; that gabapentin reduced opioid consumption and postoperative pain scores at 24 hours (p<0.001), but none has assessed the impact on quality of life. The most recent systematic review was published since this study started[25] and assessed the impact of gabapentin on length of hospital stay in eight trials which provided very low to moderate quality evidence and found no statistically significant difference in the length of hospital stay between the gabapentin and control group.

The GAP study will compare the effectiveness, cost-effectiveness and safety of gabapentin versus placebo as an adjunct to standard multimodal analgesia for the management of pain after surgery. Specific objectives are to estimate: (i) the difference between groups in length of hospital stay following surgery; (ii) the difference between groups in total opioid use, pain, adverse events and health-related quality of life (HRQoL); and (iii) the cost effectiveness of gabapentin compared with usual care.

## METHODS AND ANALYSIS
### Trial design and population
The GAP study is a multicentre, parallel group, placebo-controlled, pragmatic double-blind RCT. Patients will be recruited from three surgical specialties (cardiac, thoracic and abdominal) across several secondary care NHS centres (figure 1). A principal investigator will be appointed in each centre and clinical leads will be identified for each specialty within each centre.

GAP includes two phases: (i) phase I (12 months) involves study set-up and recruitment from two NHS secondary centres (University Hospitals Bristol and Weston NHS Foundation Trust and University Hospitals Southampton NHS Foundation Trust) with integrated monitoring of the recruitment process to maximise recruitment and adherence with the study medication;

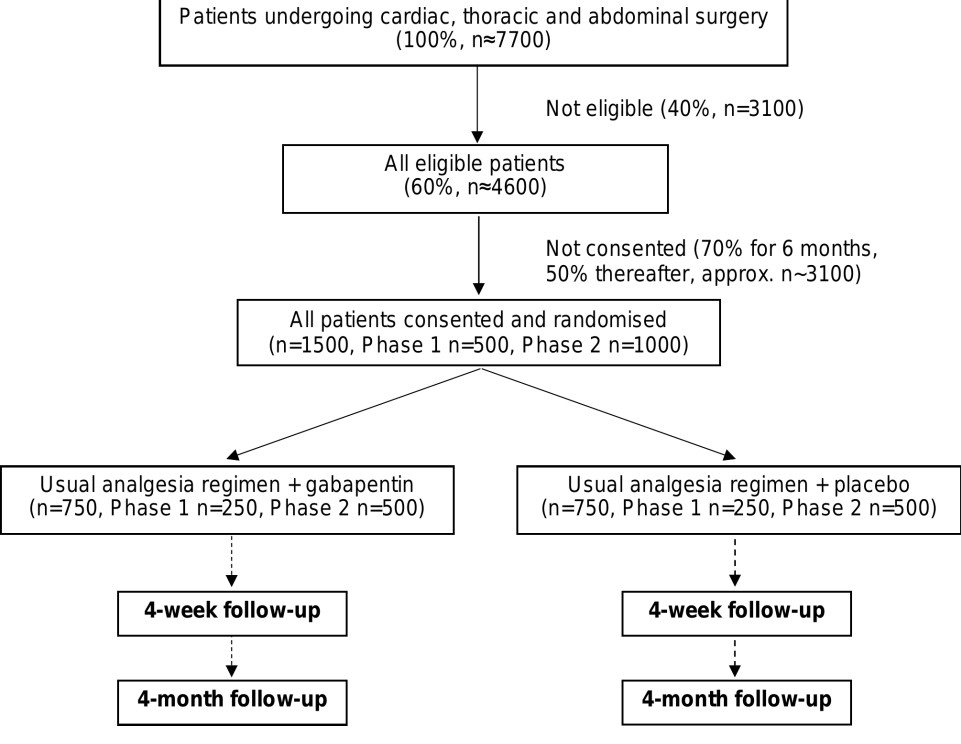

**Figure 1** Trial schema.

(ii) phase II (18 months) continued recruitment using the optimum methods established in phase I, opening additional centres (if required). Progression from phase I to phase II is contingent on demonstrating that after 9 months of recruitment in phase I, sufficient numbers of patients referred for surgery are eligible for the trial and can be enrolled to complete the main trial. Specifically:

1. At least 60% of patients undergoing surgery are considered eligible.
2. At least 50% of eligible patients consent to randomisation by 6 months of recruitment at each centre.

### Eligibility criteria
Patients will be eligible for the study if all the following apply:

1. Over 18 years of age.
2. Undergoing non-emergency surgery: (i) cardiac (surgery on the heart and great vessels performed via midline sternotomy); (ii) thoracic (open or minimal access surgery on the lungs and surrounding tissues); (iii) abdominal (open or minimal access surgery within the abdominal cavity).
3. Expected to stay in hospital at least until day 2 after surgery (day 0 is day of surgery).
4. Expected to be able to swallow during the time of the study intervention.

Patients will be excluded from the study if any of the following apply:

1. Taking antiepileptic medication(s).
2. Gabapentin allergy.
3. Already taking gabapentin or gabapentanoids.
4. Galactose intolerance, Lapp lactase deficiency or glucose-galactose malabsorption.
5. Planned epidural analgesia.
6. Intended use of any gabapentanoids in the perioperative analgesic protocol other than the study medication (this includes but is not restricted to: pregabalin, enacarbil gabapentin, 4-methylpregbalin and phenibut).
7. Known renal impairment (estimated glomerular filtration rate $<30 \, mL/min/1.73^2$).
8. Weight <50 kg.
9. Inability to provide written informed consent.
10. Unwilling to participate in follow-up.
11. Prisoners.
12. Enrolled in another clinical trial and: (i) the patient is currently taking an investigational medicinal product as part of the other trial; or (ii) coenrolment is not permitted by the other trial; or (iii) coenrolment would be burdensome for the patient.

### Patient approach and consent
Potential patients will be identified from clinic and planned operating lists and those eligible to participate will receive a patient information leaflet (PIL). Most patients will have at least 24 hours to consider participation. However, it is important to include urgent, non-emergency patients who may have less than 24 hours to consider the study, to maximise the applicability of the findings. In these circumstances, patients will only be enrolled if they confirm that they have had enough time to consider their participation.

Prior to surgery, patients will be seen by a member of the local research team who will answer any questions, confirm eligibility and receive written informed consent if the patient decides to participate. Details of all patients approached and reasons for non-participation (eg, ineligibility or patient refusal) will be documented. The patients' general practitioners will be informed of their enrolment in the study. Participants can withdraw at any time and will be treated according to standard hospital procedures. If a participant decides that they no longer wish to take part in study procedures, data collection for those procedures will cease. These participants will be asked whether they are still willing to participate in the study follow-up, if applicable.

### Interventions
The study intervention is gabapentin 600 mg given preoperatively and 600 mg/day (300 mg in the morning and 300 mg in the evening) given postoperatively for 2 days when clinically able to swallow following extubation (if applicable). The control is a placebo, taken at the same time-points as the active tablet. Both gabapentin and placebo will be administered within local multimodal analgesic regimens. The study medication (gabapentin/placebo) is manufactured, packaged and labelled in accordance with Good Manufacturing Practice and is stored at room temperature, below 25°C.

Use of any gabapentanoids other than the study medication during the study intervention period is prohibited. If the preoperative dose is administered and surgery is postponed by more than 12 hours, a second preoperative dose of study medication will be given before the rescheduled surgery. If a postoperative dose of study medication is missed by less than 6 hours, patients should be given the missed dose and continue to the next scheduled dose as per the protocol. If a dose of study medication is missed by 6 hours or more, patients should continue to the next scheduled dose and should not be given the missed dose. For patients intubated for longer than 48 hours after the end of surgery, no postoperative study medication should be administered. All other aspects of patient's care will be performed according to local practice.

### Randomisation
Randomisation will be performed after eligibility has been confirmed, using a secure internet-based randomisation system to ensure allocation concealment. Patients will be allocated in a 1:1 ratio to either gabapentin or placebo. A computer-generated allocation sequence will be prepared by an the unblinded study statistician. The random allocation will be blocked with blocks of varying size and stratified by centre and specialty, so that each specialty at each centre will have approximately equal numbers of patients allocated to placebo and gabapentin.

To maintain blinding, the randomisation system will only reveal a unique pack number, which identifies the study medication to be given.

## Blinding

Patients, their clinical care team (ie, surgeon, anaesthetist and those responsible for their postoperative care) and the research nurses will not be informed of the allocation. Patients will be made aware before entering the study that they will not be told which treatment they will receive. Doctors will prescribe 'study medication' rather than specifically gabapentin or placebo. The unique pack number provided by the randomisation system will provide the medication as specified by the predetermined randomisation list. The allocations will only be known by pharmacy and the unblinded study statistician and will not be disclosed to other members of the research team. The treatment allocation will only be unblinded if clinically indicated; for example, in the event of a suspected serious adverse reaction to the study medication, the management of which might be altered by knowledge of the allocation.

The study medication is over-encapsulated to maintain blinding. The capsules for active drug and placebo will look identical and do not have a particularly strong or unusual smell or taste, so we do not anticipate unblinding will occur due to the characteristics of the medication. Gabapentin may induce side-effects that may inadvertently unblind patients and/or clinical teams. However, given that the side effects of gabapentin (eg, drowsiness, dizziness and difficulty concentrating) are similar to those of opioids, and that patients/clinical care teams are likely to view side effects as resulting from their whole surgical and postoperative experience, it is unlikely that any patient/clinician will definitively be able to attribute a specific side effect to gabapentin. The PIL and the process of informed consent explain the uncertainty around the potential beneficial effects of gabapentin over a placebo. Therefore, in the event of inadvertent unblinding, patients should not have a strong expectation that one or other method should lead to a more favourable outcome. The success of blinding will be assessed using the Bang Blinding Index (BBI).[26]

## Outcomes

The primary outcome is length of hospital stay, from start of surgery to hospital discharge. The secondary outcomes include:

1. Acute postoperative pain assessed using the numerical rating scale completed at rest and on movement (on mobilisation, deep breathing or coughing) at 1 hour, 4 hours, 12 hours postsurgery and then two times per day until discharge.
2. Opioid consumption in the period from: (i) surgery until hospital discharge; (ii) discharge until 4 months.
3. Adverse health events in the period from: (i) randomisation to discharge; (ii) discharge until 4 months.

4. HRQoL measured using the EuroQol 5 dimension five level questionnaire (EQ-5D 5L) and Short-form (SF) 12 completed at baseline, 4 weeks and 4 months.
5. Resource use to 4 months (measured during the hospital stay, at 4 weeks and 4 months).
6. Chronic pain measured using the brief pain inventory at baseline, at 4 weeks and 4 months.

## Data collection

Screening data will be collected before consent to establish patient eligibility. The schedule of data collection outlined in table 1 will take place after consent has been received. Data will be collected onto paper data collection forms, entered onto a bespoke study database and stored on a secure server. Patient reported questionnaire data is also stored on the study database. Data for the primary outcome and most secondary outcomes will be collected during the hospital stay. Patients will be followed up at approximately 4 weeks and at 4 months for information on pain, adverse events, resource use and quality of life.

The study will end for a participant after they have completed follow-up at 4 months postsurgery. The end of the study as a whole will be after all study participants have completed follow-up, all data queries have been resolved, the database locked and the analysis completed.

## Sample size

A total of 1500 participants will be randomised to either gabapentin or placebo. The target difference in length of hospital stay was chosen to reflect the effect size that would persuade clinicians to change practice and is expressed in terms of the increase in the proportion of patients discharged at the current median time to discharge (5 days for cardiac and abdominal surgery, 3 days for thoracic surgery). This sample size will have 90% power to detect a difference of 12.5% in each specialty (ie, 50% vs 62.5%) if the number of participants per surgical stratum exceeds 376 and 80% power to detect a difference of 10% in each specialty (ie, 50% vs 60%) if the number of participants per surgical stratum exceeds 430, assuming: 5% two-sided type I error rate, 5% censoring and constant HR.

## Statistical analyses

The analyses will be conducted according to intention-to-treat and follow Consolidated Standards of Reporting Trials reporting guidelines. Randomised participants who fail to complete the course of treatment will be included in the primary analysis. All models will compare the treatment groups, will be adjusted for centre and will include a treatment by specialty interaction so the treatment effect in each surgical specialty can be quantified and compared.

The primary outcome analysis of whether there is a difference between gabapentin and placebo with respect to length of hospital stay will use Cox proportional hazards regression. Those participants who die before discharge will be censored at the longest recorded length of stay

**Table 1** Schedule of data collection

| Data item | Prerandomisation | Presurgery | Intra operative | Postsurgery (until discharge) | Discharge | 4 weeks post surgery | 4 months post surgery |
|---|---|---|---|---|---|---|---|
| Sociodemographic details | ✓ | | | | | | |
| Comorbidities | ✓ | | | | | | |
| Routine clinical measures | ✓ | | | | ✓ | | |
| Resource use schedule | | | | | ✓ | ✓ | ✓ |
| SF-12 | ✓ | | | | | ✓ | ✓ |
| EQ-5D 5L | ✓ | | | | | ✓ | ✓ |
| BPI | ✓ | | | | | ✓ | ✓ |
| NRS pain score | ✓* | | | ✓* | ✓ | | |
| Study medication | | ✓ | | ✓† | | | |
| Opioid use | ✓ | | ✓ | ✓ | ✓ | ✓ | ✓ |
| Adverse events | | | | ✓ | ✓ | | |
| Serious adverse events | | | | ✓ | ✓ | ✓ | ✓ |

*Routinely collected NRS pain scores as close as possible to the following time points may be used: prerandomisation, 1 hour, 4 hours, 12 hours postsurgery and two times per day postsurgery until discharge. NRS pain assessments will not be possible in intubated patients.
†Study medication given morning and evening for 2 days following extubation (where applicable).
BPI, brief pain inventory; NRS, numerical rating scale.

for that specialty, as this is computationally equivalent to competing risk methodology in this setting.

Opioid consumption, pain scores and HRQoL outcomes will all be analysed using mixed regression models, adjusted for baseline measures where appropriate. Changes in treatment effect with time will be assessed by adding a treatment by time interaction to the model and comparing models using a likelihood ratio test. Deaths will be accounted for by modelling HRQoL and survival jointly. Model fit will be assessed and alternative models and/or transformations (eg, to induce normality) will be explored where appropriate. Safety will be assessed by summarising the number and proportion of participants reporting serious and non-serious adverse events and will be reported to the Data Monitoring and Safety Committee (DMSC) on a regular basis.

The health economic evaluation will compare the costs and effects of gabapentin compared with placebo for the management of pain after major surgery. The within-trial cost-effectiveness analysis will be undertaken from an NHS and personal social services perspective, with a 4-month time horizon from the day of surgery. Effects will be measured using quality-adjusted life years (QALYs), estimated using EQ-5D 5L.[27 28] Costs will include medication costs and those related to inpatient stay. Established guidelines as set out by the National Institute for Health and Care Excellence[29] will be followed for the economic evaluation. The incremental cost-effectiveness ratio will be calculated from the average costs and QALYs in each trial group to produce an incremental cost per QALY of gabapentin compared with placebo.[30]

Exploratory subgroup analyses are planned to explore the primary and secondary outcomes in terms of type of surgery (open/minimal access).

### Data handling, storage and sharing
Data will be stored in a bespoke database hosted on the NHS network. Access to the database will be via a secure password-protected web-interface. All study documentation will be retained in a secure location during the study and for 15 years after the end of the study, when all patient identifiable paper records will be destroyed by confidential means. Medical records documenting study related information will be identified by a label bearing the name and duration of the study. In compliance with the Medical Research Council Policy on Data Sharing, relevant 'meta'-data about the study and the full dataset, but without any participant identifiers other than the unique participant identifier, will be held indefinitely on a University of Bristol server. A secure electronic 'key' with a unique participant identifier, and key personal identifiers will also be held indefinitely, but in a separate file and in a physically different location (NHS hospital server). These will be retained because of the potential for the raw data to be used subsequently for secondary research.

### Risk of bias
The following key features have been incorporated into the study to minimise the risk of bias:
1. Selection/allocation bias arising from the randomisation process will be prevented by using computer-generated concealed randomisation. Allocation lists

prepared by an unblinded statistician will be stratified by centre and specialty to minimise confounding. Participants will be randomised after eligibility is confirmed.

2. Performance bias arising from deviations from intended interventions will be minimised by blinding all participants, clinicians and other hospital staff caring for participants and members of the research team (apart from the study unblinded statistician) to participants' allocation. The success of blinding will be assessed by asking participants and research nurses responsible for participant care and data collection to complete the BBI at the point of discharge from hospital. Participants will complete the BBI again 4 months after surgery.[26] Performance bias will also be minimised by administering the study medication according to standard protocols and by predefining all other study procedures and applying these to all participants in the same way. Adherence to all aspects of the protocol will be monitored.

3. Detection bias arising from differences in how the outcome is measured will be minimised by blinding all individuals assessing outcomes, assessing the success of blinding and providing clear unambiguous definitions for each outcome measure.

4. Attrition bias arising from missing outcome data will be minimised by (i) maintaining contact with participants throughout the duration of the study to maximise the proportion of participants for whom all outcome data are available, (ii) implementing measures to promote adherence (eg, training for staff administering the intervention, posters to remind the care team of patient study participation) and (iii) documenting non-adherence to the allocated treatment. The data will also be analysed by intention-to-treat. In estimating the target sample size, loss to follow-up has not been allowed for as the primary outcome is time to hospital discharge and the follow-up period is short (4 months). However, attention will be paid to keeping in touch with participants and maximising retention up to 4 months.

5. Reporting bias will be minimised by having prespecified outcomes and a prespecified analysis plan.

## Patient and public involvement

Patient and public involvement (PPI) input was sought at the study design phase from relevant surgical PPI groups: the National Institute of Health Reasearch Bristol Biomedical Research Centre (Nutrition) colorectal PPI group, the Royal Brompton Hospital Cancer Consortia PPI group and patients who underwent cardiac surgery at the Bristol Heart Institute. GAP also includes a patient co-applicant. All PPI groups and the patient co-applicant unanimously agreed that the study was important and welcomed treatments that might reduce the amount of opioid drugs patients receive, and their associated side effects, after surgery. PPI groups provided feedback on the study intervention and outcome data collection (eg,

pain scores and questionnaires), which informed the study design.

PPI engagement will continue during study implementation, including writing and designing participant-facing documents and outlining the participant follow-up schedule. The GAP study Trial Steering Committee (TSC) includes two public members who regularly review study progress.

PPI groups will continue to help with all aspects of the study, including preparing lay results summaries for dissemination to participants and other patient groups in order to maximise public awareness of the findings.

## Ethics and dissemination

The study received Research Ethics Committee (REC) approval from Yorkshire and the Humber—Sheffield REC in November 2017, Medicines and Healthcare Regulatory Agency approval in December 2017 and Health Research Authority (HRA) approval in January 2018.

The study is sponsored by University Hospitals Bristol and Weston NHS Foundation Trust (www.uhbristol.nhs.uk/research-innovation/) and is coordinated by the Bristol Trials Centre, Clinical Trials and Evaluation Unit, (BTC (CTEU)), a UK Clinical Research Collaboration registered Clinical Trials Unit (reference 11). The TSC is made up of representatives from the GAP study team and independent members approved by the funder. The DMSC consists of an independent medical statistician and medical experts in this field approved by the funder. The TSC and DMSC meet as frequently as they feel is necessary, usually at least once a year.

## Changes to the protocol since REC/HRA approval

Following REC and HRA approval several changes have been made to the study protocol, as follows: (i) safety reporting requirement updates; (ii) reference safety drug information updates; (iii) clarifications about the level of care provided to study participants; (iv) clarifications that patients must be expected to be able to swallow during the time of the study intervention to be eligible; (v) clarification that the first postoperative dose should only be administered if patients are clinically able to swallow; (vi) study medication packs contain six capsules instead of eight (to minimise the chance of participants receiving more study medication doses than intended); (vii) permitting eligibility and prescription sign off by non-doctor clinicians (eg, nurse practitioners); (viii) provision of optional patient diaries; (ix) opening to recruitment from more centres; and (x) study team contact detail updates. Protocol version 8.0 (dated 03 December 2019) is currently in use.

## Dissemination of findings

Findings will be disseminated to participating hospitals and to the academic community through peer-reviewed publications and presentation at national and international meetings. Findings will also be shared with study participants who express a wish to receive study results

through patient organisations, leaflets and newsletters. Study updates are regularly provided to the study team, participants and members of the public though emails, newsletters, magazine articles and social media.

## DISCUSSION

The study design, involving three surgical specialties, was chosen because it is efficient and maximises the value of the research for the NHS. The inclusion of different surgical specialties reflects current clinical practice (gabapentin is prescribed to patients undergoing different surgical procedures) and should make the trial results generalisable in the NHS. The study opened to recruitment on 12 April 2018 and is currently recruiting in six centres. To date, 853 patients have been recruited (469 cardiac, 221 thoracic, 163 abdominal). The progression criteria were met and approvals to progress to phase II were received on 04 March 2019. GAP has proven more difficult to deliver than anticipated for a study which was perceived to have a straightforward intervention. Patient eligibility and patient willingness to participate have not been a limit to recruitment. Some of the challenges of delivering the study include (i) higher than expected training requirements to integrate administration of study medication into routine clinical practice in all specialties at participating centres, (ii) difficulties of using multiple clinical prescribing systems (electronic and paper) which are not linked and require multiple manual updates for a single in-hospital patient stay, (iii) higher than anticipated research team resource required to meet regulatory requirements (eg, obtaining clinician eligibility sign off, often during unsocial hours, or additional requirements following the reclassification of gabapentin as a schedule three controlled drug in April 2019); and (iv) regulatory structures that do not permit the study to have a designated principal investigator for each specialty in a centre. This is particularly challenging when patients are under the care of different clinical teams that are administratively and geographically separate. Further details about the challenges of delivering the GAP study in an NHS setting will be reported elsewhere. This study highlights that while the design is methodologically attractive, the current regulatory structures and NHS systems make implementation suboptimal.

## Author affiliations
[1]Bristol Trials Centre, Clinical Trials and Evaluation Unit, Bristol Medical School, University of Bristol, Bristol, UK
[2]Department of Surgery, University Hospitals Bristol and Weston NHS Foundation Trust, Bristol, UK
[3]Department of Surgery, University Hospital Southampton NHS Foundation Trust, Southampton, UK
[4]Department of Anaesthesia, University Hospitals Bristol and Weston NHS Foundation Trust, Bristol, UK
[5]Department of Anaesthesia, University Hospital Southampton NHS FoundationTrust, Southampton, UK
[6]Acute, Critical & Perioperative Care Research Group, NIHR Biomedical Research Centre, University Hospital Southampton NHS Foundation Trust/University of Southampton, Southampton, UK
[7]Health Economics Research Centre, University of Oxford, Oxford, UK
[8]Bristol Medical School, University of Bristol, Bristol, UK

**Acknowledgements** The GAP study is sponsored by University Hospitals Bristol and Weston NHS Foundation Trust. This study was designed and delivered in collaboration with the BTC (CTEU), a UKCRC registered clinical trials unit which is in receipt of National Institute for Health Research Clinical Trial Unit support funding. The authors would like to thank all the research and clinical team members involved in recruitment,coordination, delivery of the intervention, data collection and data entry for this study.

**Contributors** SB is involved in conducting the trial and assembled the manuscript from the trial protocol. MP identified the funding opportunity and designed the trial with statistical input from CR. CR is the non-clinical lead and the methodology/statistics lead for the trial. BG is the chief investigator and NC is the clinical pain lead for the trial. LCo drafted the statistical analysis plan. SW is the health economics lead and EAS the health economist working on the trial. SB and EAS designed the data collection for the health economics element of the trial. LCu provides senior trial management oversight and advice. HM, SEdJ and JL have assisted with the set-up and delivery of the trial. RA, AA, GC, ME, NG and MM are participating clinicians in the trial. All authors have been involved in preparation of the study protocol and have read and approved the final manuscript.

**Funding** This study is funded by the National Institute for Health Research Health Technology Assessment Programme (15/101/16). The views and opinions expressed therein arethose of the author(s) and do not necessarily reflect those of the National Institute of Health Research, NHS, or the Department of Health and Social Care.

**Competing interests** No competing interests are declared. At the time that the work was carried out, GC was employed by the University Hospitals Bristol NHS Foundation Trust, Bristol, UK. GC is currently the Medical Director Johnson and Johnson Medical Devices UK and Ireland. GC has no competing interests to declare.

**Patient and public involvement** Patients and/or the public were involved in the design, or conduct, or reporting, or dissemination plans of this research. Refer to the Methods section for further details.

**Patient consent for publication** Not required.

**Provenance and peer review** Not commissioned; externally peer reviewed.

**ORCID iDs**
Sarah Baos http://orcid.org/0000-0003-2039-8768
Chris A Rogers http://orcid.org/0000-0002-9624-2615
Reyad Abbadi http://orcid.org/0000-0002-5334-5227
Aiman Alzetani http://orcid.org/0000-0002-3373-6714
Nilesh Chauhan http://orcid.org/0000-0002-2307-9204
Laura Collett http://orcid.org/0000-0002-6681-2146
Lucy Culliford http://orcid.org/0000-0002-9255-6617
Samantha E de Jesus http://orcid.org/0000-0002-9905-2737
Mark Edwards http://orcid.org/0000-0002-5048-1784
Holly McKeon http://orcid.org/0000-0002-0220-503X
Elizabeth A Stokes http://orcid.org/0000-0002-4179-1369
Sarah Wordsworth http://orcid.org/0000-0002-2361-3040
Ben Gibbison http://orcid.org/0000-0003-3635-6212
Maria Pufulete http://orcid.org/0000-0002-1775-1949

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
