## [Reviewer comments · BMJ Open]

ARTICLE DETAILS

TITLE (PROVISIONAL)	Effectiveness, cost effectiveness and safety of gabapentin versus placebo as an adjunct to multimodal pain regimens in surgical patients: Protocol of a placebo controlled randomised controlled trial with blinding (GAP study)
AUTHORS	Baos, Sarah; Rogers, Chris; Abbadi, Reyad; Alzetani, Aiman; Casali, Gianluca; Chauhan, Nilesh; Collett, Laura; Culliford, Lucy; de Jesus, Samantha; Edwards, Mark; Goddard, Nicholas; Lamb, Jennifer; McKeon, Holly; Molyneux, Mat; Stokes, Elizabeth; Wordsworth, Sarah; Gibbison, Ben; Pufulete, Maria

VERSION 1 – REVIEW

REVIEWER	Lucyna Tomaszek Department of Nursing, Faculty of Medicine and Health Sciences, Andrzej Frycz Modrzewski Krakow University, Krakow, Poland
REVIEW RETURNED	01-Aug-2020

GENERAL COMMENTS	Thank you for the opportunity to review this well-presented protocol describing the use of gabapentin as a single preoperative oral dose and as multiple postoperative doses in the treatment of post-operative pain in adults. Protocol manuscript is written according to the SPIRIT (Standard Protocol Items for Randomized Trials). In introduction Authors enough explained the rationale for the study based on appropriate literature, including relevant systematic reviews. The aims of the study are clear and the methodology appropriate to fulfilling the aims of the study. I have a few minor suggestions to strengthen the manuscript. Title: please, adjust title to the purpose of the research or vice versa (e.g. title „Effectiveness, safety and cost-effectiveness of gabapentin versus placebo.....; purpose: „The GAP study will compare the effectiveness, safety and cost-effectiveness of gabapentin as an adjunct.....“) Page 4, lines 39-43: References are needed. Page11, lines 45-50. It is not clear what pain scale is used, NRS or VAS (ISRCTN63614165 https://doi.org/10.1186/ISRCTN63614165; „Acute post-operative pain is assessed using the visual analogue scale (VAS) completed at 1, 4 and 12 hours post-surgery and then twice daily to discharge”). Please, explain how pain intensity is measured (e. g. at rest, during deep breathing, and coughing) and add information about a therapeutic goal of pain control (e. g. the goal for effective pain management was to maintain patients’ pain intensity at 3/10 at rest). References. Capital letters should be used for each word of the
--

	name of the journal (e.g. Current Medical Research and Opinion) and cited journals should be abbreviated (e.g. Curr. Med. Res. Opin.).
--	--

REVIEWER	Ian Gilron Queen's University
REVIEW RETURNED	28-Sep-2020

GENERAL COMMENTS	The authors present the protocol for a multicenter, double-blind, randomized controlled, 2-arm trial (n=750 participants per arm) comparing gabapentin (600mg just before surgery and 600mg/day for 2 days after surgery) to identically appearing inert placebo in adult participants undergoing different types of major surgery (cardiac, thoracic or abdominal). The primary outcome is length of hospital stay following surgery. Secondary outcomes include pain, total opioid use, adverse health events, health related quality of life and costs. Given what has been recognized to be a modest, or clinically insignificant, effect of gabapentin on postoperative pain and/or opioid use – that has been previously described in several meta-analyses of dozens of trials – one might question the cost and effort invested into this trial, particularly if a negative or equivocal trial result does not sufficiently convince clinicians to stop using this drug for postoperative pain. In describing previous evidence from perioperative studies of gabapentin, the authors appropriately acknowledge the heterogeneity of studies – with respect to surgical procedure, gabapentin dose and duration and other factors. Unfortunately, the trial population being proposed here will be similarly heterogeneous regarding surgical procedure. Although the proposed trial is larger than many previous gabapentin trials, the size of 750/arm may still not be large enough for any meaningful subgroup analyses by surgical procedure. Selecting postoperative length of hospital stay as a primary outcome is problematic for a number of reasons: 1) gabapentin is generally not prescribed to patients to hasten hospital discharge but rather to reduce pain; 2) even if gabapentin can substantially reduce postoperative pain and/or opioid use, this pain reduction may be necessary – but not sufficient – to hasten hospital discharge; 3) there are other factors not affected by gabapentin administration that could have an important impact on hospital stay (e.g. surgical site infections, other complications). The potential for any ‘disconnect’ between hospital stay and pain reduction could confuse the conclusions of the trial. For example, if the trial result is negative, i.e. gabapentin does not affect hospital stay, however, gabapentin does significantly reduce pain, this would send a mixed message to clinicians. The Introduction for the manuscript should cite a recent meta-analysis (Verret et al, Perioperative Use of Gabapentinoids for the Management of Postoperative Acute Pain: A Systematic Review and Meta-analysis. Anesthesiology 2020;133:265).
--

VERSION 1 – AUTHOR RESPONSE

Response to Reviewer 1

- Thank you for the opportunity to review this well-presented protocol describing the use of gabapentin as a single preoperative oral dose and as multiple postoperative doses in the treatment of post-operative pain in adults. Protocol manuscript is written according to the SPIRIT (Standard Protocol Items for Randomized Trials). In introduction Authors enough explained the rationale for the study based on appropriate literature, including relevant systematic reviews. The aims of the study are clear and the methodology appropriate to fulfilling the aims of the study. I have a few minor suggestions to strengthen the manuscript.
- We would like to thank the reviewer for their input.
- Title: please, adjust title to the purpose of the research or vice versa (e.g. title „Effectiveness, safety and cost-effectiveness of gabapentin versus placebo.....; purpose: ,The GAP study will compare the effectiveness, safety and cost-effectiveness of gabapentin as an adjunct.....))
- We have updated the purpose on page 6 to match the title as suggested.
- Page 4, lines 39-43: References are needed.
- We have added a reference to page 4 as requested.
- Page11, lines 45-50. It is not clear what pain scale is used, NRS or VAS (ISRCTN63614165 <https://doi.org/10.1186/ISRCTN63614165>; „Acute post-operative pain is assessed using the visual analogue scale (VAS) completed at 1, 4 and 12 hours post-surgery and then twice daily to discharge”).
- Thank you for noticing this discrepancy. We can confirm that acute pain is measured using the numerical rating scale (NRS). This error has been corrected on the ISRCTN record.
- Please, explain how pain intensity is measured (e. g. at rest, during deep breathing, and coughing) and add information about a therapeutic goal of pain control (e. g. the goal for effective pain management was to maintain patients’ pain intensity at 3/10 at rest).
- We have clarified on page 12 that pain scores are collected at rest and on movement (on mobilisation, deep breathing, or coughing). This is a pragmatic study and as such we do not stipulate what the therapeutic goal for pain control should be.
- References. Capital letters should be used for each word of the name of the journal (e.g. Current Medical Research and Opinion) and cited journals should be abbreviated (e.g. Curr. Med. Res. Opin.).
- We have updated the formatting of the references to comply with BMJOpen guidelines.

Response to Reviewer 2

- Given what has been recognized to be a modest, or clinically insignificant, effect of gabapentin on postoperative pain and/or opioid use – that has been previously described in several meta-analyses of dozens of trials – one might question the cost and effort invested into this trial, particularly if a negative or equivocal trial result does not sufficiently convince clinicians to stop using this drug for postoperative pain.
- This study was commissioned by the UK National Institute of Health Research (NIHR) to provide a definitive answer to the questions regarding effectiveness, cost-effectiveness and safety of gabapentin when used as an analgesic after surgery. We agree that a negative or equivocal result

with respect to 'length of hospital stay' will mean that clinicians will continue to use gabapentin, particularly if it is shown to reduce pain and result in no adverse effects.

- In describing previous evidence from perioperative studies of gabapentin, the authors appropriately acknowledge the heterogeneity of studies – with respect to surgical procedure, gabapentin dose and duration and other factors. Unfortunately, the trial population being proposed here will be similarly heterogeneous regarding surgical procedure. Although the proposed trial is larger than many previous gabapentin trials, the size of 750/arm may still not be large enough for any meaningful subgroup analyses by surgical procedure.

- The rationale for including multiple surgical specialities is that the mechanism of the body's response to trauma/injury caused by surgery and indeed the treatment for acute postoperative pain is the same regardless of the type of surgery a patient undergoes. In support of this, previous meta-analyses show a similar effect size in terms of pain reduction regardless of surgical specialty, which explains why most meta-analyses combine all surgical populations. Unlike most RCTs, this study is powered for a subgroup analysis by surgical specialty. The minimum number of patients required per specialty to detect the pre-specified target difference (50% versus 62.5% discharged by the current median time to discharge, hazard ratio 1.41) was set at 376 per specialty and was inflated to 500 to allow for intervention non-compliance).

- Selecting postoperative length of hospital stay as a primary outcome is problematic for a number of reasons:

- 1) gabapentin is generally not prescribed to patients to hasten hospital discharge but rather to reduce pain;

- At the time we designed the study, all meta-analyses showed that gabapentin reduced pain and opioid use. What was unclear was whether reduced pain and reduced opioid use translates into a faster recovery from surgery and reduced costs for healthcare providers. Length of hospital stay is a surrogate marker for complications and morbidity after surgery; outcomes important to patients and healthcare providers.

- 2) even if gabapentin can substantially reduce postoperative pain and/or opioid use, this pain reduction may be necessary – but not sufficient – to hasten hospital discharge;

- We agree. This is the central question which the trial is aiming to answer.

- 3) there are other factors not affected by gabapentin administration that could have an important impact on hospital stay (e.g. surgical site infections, other complications).

- We agree that other factors are likely to affect length of hospital stay. However, randomisation should balance these across the two groups, so that if we observe a reduction in length of stay it is likely to be due to the study intervention. This is especially true given that our study design (double blind and placebo controlled) puts the trial at low risk of bias.

- The potential for any 'disconnect' between hospital stay and pain reduction could confuse the conclusions of the trial. For example, if the trial result is negative, i.e. gabapentin does not affect hospital stay, however, gabapentin does significantly reduce pain, this would send a mixed message to clinicians.

- We do not think that a disconnect between length of hospital stay and pain reduction would confuse the conclusions of the trial. If gabapentin results in a clinically meaningful pain reduction with a good safety profile, but does not affect length of hospital stay, clinicians may continue to prescribe gabapentin for pain management as part of a multimodal analgesic regimen, knowing that it is safe to do so.

- The Introduction for the manuscript should cite a recent meta-analysis (Verret et al, Perioperative Use of Gabapentinoids for the Management of Postoperative Acute Pain: A Systematic Review and

Meta-analysis. Anesthesiology 2020;133:265).

- This reference has been added to page 5.

VERSION 2 – REVIEW

REVIEWER	Ian Gilron Queen's University
REVIEW RETURNED	15-Oct-2020
GENERAL COMMENTS	The authors have revised the manuscript appropriately in response to previous reviewers' comments. No further comments here.